# Immune Checkpoint Blockade for Advanced NSCLC: A New Landscape for Elderly Patients

**DOI:** 10.3390/ijms20092258

**Published:** 2019-05-07

**Authors:** Fabio Perrotta, Danilo Rocco, Fabiana Vitiello, Raffaele De Palma, Germano Guerra, Antonio De Luca, Neal Navani, Andrea Bianco

**Affiliations:** 1Department of Medicine and Health Sciences “V. Tiberio”, University of Molise, 86100 Campobasso, Italy; germano.guerra@unimol.it; 2Pneumo-Oncology Unit, A.O. dei Colli “Monaldi Hospital”, 80131 Naples, Italy; danilorocc@yahoo.it (D.R.); f.vitiello@aodeicolli.it (F.V.); 3Department of Precision Medicine, University of Campania “L. Vanvitelli”, 80131 Naples, Italy; raffaele.depalma@unicampania.it; 4Department of Mental and Physical Health and Preventive Medicine, Section of Human Anatomy, University of Campania “L. Vanvitelli”, 80131 Naples, Italy; antonio.deluca@unicampania.it; 5Lungs for Living Research Centre, UCL Respiratory and Department of Thoracic Medicine, University College London Hospital, London WC1E6JF, UK; n.navani@ucl.ac.uk; 6Department of Translational Medical Sciences, University of Campania “L. Vanvitelli”, 80131 Naples, Italy; andrea.bianco@unicampania.it

**Keywords:** NSCLC, Immune checkpoint inhibitors, PD-1/PD-L1, Elderly patient, biological big data, precision oncology

## Abstract

The therapeutic scenario for elderly patients with advanced NSCLC has been limited to radiotherapy and chemotherapy. Recently, a novel therapeutic approach based on targeting the immune-checkpoints has showed noteworthy results in advanced NSCLC. PD1/PD-L1 pathway is co-opted by tumor cells through the expression of PD-L1 on the tumor cell surface and on cells within the microenvironment, leading to suppression of anti-tumor cytolytic T-cell activity by the tumor. The success of immune-checkpoints inhibitors in clinical trials led to rapid approval by the FDA and EMA. Currently, data regarding efficacy and safety of ICIs in older subjects is limited by the poor number of elderly recruited in clinical trials. Careful assessment and management of comorbidities is essential to achieve better outcomes and limit the immune related adverse events in elderly NSCLC patients.

## 1. Introduction

Lung cancer represents the major cause of cancer death worldwide and non-small cell lung cancer (NSCLC) represents more than 80% of newly diagnosed lung cancer [1]. NSCLC is predominant among older adults; an epidemiologic study showed that 47% of lung cancers are diagnosed in subjects aged more than 70 years and 15% are diagnosed in patients older than 80 [1]. Despite advances in understanding about interaction between risks factors and molecular oncogenic pathways [2,3,4], the exact relation between aging and the increased prevalence of lung cancer in the elderly population remains a matter for debate [5,6,7,8,9,10]. Immunosenescence is a highly dynamic process of remodeling and adaptation that lead to changes in lymphocyte subpopulations and lymphocyte differentiation, which may have different roles in oncogenesis [11]. Shifts in adaptive immunity, including a decline in the number of naïve T cells, rise of memory CD4 T cells and higher concentrations of inflammatory cytokines have been described [12,13]. Elderly people with malignant tumors are expected to rise due to global aging of the population coupled with early detection of lung cancer. Some researchers have investigated screening programs in older population reporting early diagnosis to be feasible in older subjects [14,15,16].

Elderly NSCLC patients have the lowest rates of anti-cancer treatment and the poorest prognosis [17,18]. Although some evidence suggests that in the early stages minimally invasive surgery remains the mainstay of the treatment [19,20,21,22,23], for patients with advanced disease, therapy has been limited for some years to chemotherapy, targeted therapies and radiotherapy. Several factors, including limited physiologic reserve, reduced therapeutic options and increased risks of iatrogenic toxicities as well as age-related comorbidities influence mortality in elderly patients [24]. However, during the last few years a growing body of research based on enhancing the antigen specific T-cell response after immune checkpoint inhibition has shown promising results in clinical practice [25,26]. Cancer cells may express checkpoint blockade molecules disrupting immune recognition pathways thus inducing enhanced cell survival. Molecules involved in regulating the immune system include: PD-1 (programmed cell death protein-1) or its’ ligand (PDL-1) and CTLA-4 (cytotoxic T-lymphocyte-associated antigen-4). Binding between PD-1 and its ligand, which may be expressed on the cancer cells surface, inactivates the T cell response. Immune checkpoint inhibitors (ICI’s) block PD-1 (Nivolumab and Pembrolizumab) or PD-L1 (Durvalumab, Atezolizumab, Avelumab) and have already been approved for treatment of advanced NSCLC or are awaiting approval. Unfortunately, the elderly population is generally underrepresented in NSCLC clinical trials and most of the evidence arises from selected study populations. Data reported from the Southwest Oncology Group reported that cancer clinical trials results might not be generalizable to the elderly population as only a quarter of trial participants are 65 years or older [27,28]. Immunosenescence represents a key point in determining response to immunotherapy in the elderly. Although different data have been reported in literature, it is known that expression of PD1 and CTLA-4 changes with age. Age depending dynamics regulating expression of inhibitory molecules on T cells are not yet understood; conflicting results reported may be due to differences between animal models and patients. Furthermore, it is well known that modulation of PD1 and CTLA-4 due to exhaustion of T effector cells and decline of Treg cells are observed in senescence. These data suggest that elderly patients may have a peculiar response to therapy targeting immune checkpoints [29,30]. This review examines the principal aspects of immunosenescence, metagenomic data on gut microbioma and their implications for immune checkpoint inhibition in NSCLC and the results of immunotherapy clinical trials in the elderly population.

## 2. Immunosenescence and PD-1/PD-L1 Pathways

Aging is associated with decline and impairment of immune function. This condition, defined as immunosenescence, may contribute to the increased incidence and development of cancer amongst the elderly and may affect the efficacy and the toxicity of cancer treatment, including treatment with immune checkpoint inhibitors [31]. Several studies have shown that aging is associated with impaired function and production of lymphocyte [32,33,34] coupled with changes of hematopoietic stem cells (Figure 1) [35,36].

The Naïve T cell compartment is significantly influenced by age-related decline of the immune system. While the CD8+ naïve T cell pool decreases, the proportion of CD4+ naïve T cells appears to remain stable [37]. However, while numbers of CD4+ naïve T cells appear to be preserved, there appears to be a defect in T cell function [38] and in T cell receptor (TCR) repertoire, resulting in an impaired ability to recognize new antigens [39]. Furthermore, the decreased expression of costimulatory protein CD28 and CD40 ligand by CD4+ T cell contributes to the impairment of CD4+ T cell function [40] that secondarily impacts also CD8+ cell activity.

The CD8+ T cells are the principal elements involved in the PD-1/PDL-1 pathway and are mainly affected by age-related decline. Some studies have showed that CD8+ cells have an oligoclonal TCR repertoire and low CD28 expression [41,42,43], reduced proliferative capacity and increased sensitivity to apoptotic signals [44]. The accumulation of CD28- CD8+ T cells in older adults leads not only to decreased immune activation [40,42], but seems to have suppressive activity and increases in tumors [45]. Cytotoxic T cells in the elderly have an increased expression of CD57, a marker of senescence, that lead to less anti-tumoral immunogenic activity [46] and lower level of perforine and granzyme, essential for cytotoxic activity [47]. Moreover, some studies demonstrate an increased expression of PD-1 on the surface of T cells but its blockade does not restore T cell activity to the same level of younger people [48,49].

PD-1 is also expressed on the surface of other cells such as B cells, Natural Killer (NK) cells, Dendritic cells (DCs), myeloid cells and therefore it is likely that anti-PD-1 antibodies can also stimulate anti-tumor immunity increasing antibody production, activating NK cells or reducing immunosupressive activity of “myeloid derived suppressor cells” (MDSCs) [50,51,52]. MDCSs are a subset of myeloid cells that are in the tumor microenvironment and can express PD-L1 on their surface, contributing to suppression of T cell function. In elderly patients, the number of MDSCs increases both in tumor stroma and in the circulation [36,53] and in older mice this increase corresponds to a greater susceptibility to lung cancer [54]. It appears that age-related immune molecular changes have a great impact on cancerogenesis and, at the same time, may influence the efficacy and the activity of checkpoint inhibitors, including PD-1 and PD-L1 inhibitors [55]. Conversely, the tumor mutational burden (TMB), defined as the total number of somatic mutations in a defined region of a tumour genome, a potential predictor of response to ICIs, appears to increase with age [56].

## 3. ICI’S for Advanced NSCLC in Elderly Population

Several ICIs have been approved by the European Agency of Medicine (EMA) and the Food and Drug Administration (FDA) in the treatment of NSCLC. Pembrolizumab is authorized in first line for advanced non-squamous or squamous NSCLC without driver genes mutations (EGFR/ALK) expressing PD-L1 ≥ 50% and in the second line for advanced non-squamous or squamous NSCLC expressing PD-L1 ≥ 1%. The use of Nivolumab and Atezolizumab is currently authorized for advanced non-squamous or squamous NSCLC with negative or unknown PD-L1 expression after at least one previous chemotherapy regimen. Durvalumab has been approved in Europe as consolidation therapy after chemo-RT in in unresectable stage III NSCLC that express PD-L1 at levels ≥1%. Table 1 summarizes the effects in different age group populations of ICIs approved by regulatory agencies for NSCLC treatment.

### 3.1. Pembrolizumab

Pembrolizumab is a humanized IgG4 antibody that targets the PD-1 protein. Two registration studies authorize use in first or second line, depending on the percentage of PD-L1 expression. In the KEYNOTE-010 [57], a Phase II/III randomized trial, 1034 patients already treated with a previous chemotherapy regimen (70% one line, 30% with two or more chemotherapeutic lines) expressing PD-L1 on at least 1% of tumor cells, were enrolled to establish the role of pembrolizumab (at two different doses of 2 and 10 mg/kg) as a second-line therapy compared to docetaxel. The cohort of patients had a mean age of 63 years (range 54–69). Pembrolizumab reported a significant OS improvement regardless of the dose administered (HR: 0.71, *p* = 0.0008 to 2 mg/kg, HR: 0.61, *p* < 0.0001 to 10 mg/kg). Median PFS was similar across the groups (3.9 months with pembrolizumab 2 mg/kg, 4.0 months with pembrolizumab 10 mg/kg and docetaxel). Among patients with strongly positive PD-L1 (defined as the expression of PD-L1 in at least 50% of tumor cells), OS and PFS were significantly higher with pembrolizumab, regardless of the dose, than docetaxel (median OS: 14.9 and 17.3 months with pembrolizumab at 2 and 10 mg/kg, respectively, compared with 8.2 months with docetaxel; median PFS: 5.0 and 5.2 months with pembrolizumab at 2 and 10 mg/kg, respectively, compared to 4.1 months with docetaxel). In KEYNOTE 010, a subgroup analysis by defined age reported a significant reduction in the risk of death by 24% in 429 patients aged ≥ 65 years [HR: 0.76 (95% CI, 0.57–1.02). In KEYNOTE 024 60 Pembrolizumab as first line therapy showed an improvement in PFS of approximately 4 months compared to standard chemotherapy (10.3 vs. 6.0 months, HR 0.50) in NSCLC patients with PD-L1 expression more than 50%. This benefit was reconfirmed in all subgroups examined regardless of the patients age.

### 3.2. Nivolumab

Nivolumab is a fully human IgG4 anti PD-1 antibody that controls the immune checkpoint and stops PD-1-mediated signaling and restores anti-tumor immunity. Nivolumab (3 mg/kg every 2 weeks) as single agent was compared with docetaxel as second-line treatment in squamous NSCLC (CheckMate-017) [58] and non-squamous NSCLC (Checkmate-057) [59], with OS as the primary endpoint in these two phase III studies. In Checkmate-017, the study population consisted of 272 patients with a mean age of 63 (range 39–85). The study revealed longer OS and PFS in nivolumab group than with docetaxel group regardless PD-L1 expression levels of the tumor (TPS 1%, 5% or 10%). An analysis of the subgroups shows improvement in OS and PFS in patients aged 65 to 75 years (HR of 0.56 and 0.51, respectively), but not in patients >75 years of age (OS HR: 1.85; PFS HR 1.76), although the small sample size (29 patients) does not allow statistically significant conclusions. In Checkmate-057, the study population consisted of a cohort of 582 patients with a median age of 62 (range 21–85). In this study the OS was significantly longer with nivolumab than with docetaxel, with a 27% lower death risk in the nivolumab group (HR: 0.73, 96% CI, 0.59–0.89, *p* = 0.002), a benefit in terms of median survival of 2.8 months (median OS 12.2 vs. 9.4 months) and a 1-year OS rate of 51% compared to 39% with docetaxel. Although non-squamous PFS did not favor nivolumab compared to docetaxel (median 2.3 vs. 4.2 months, respectively), the PFS rate at 1 year was higher with nivolumab compared to chemotherapy (19% and 8%, respectively). In contrast to CheckMate-017, the magnitude of benefit across all efficacy endpoints seemed to be related to PD-L1 expression. An analysis of the subgroups shows an improvement in OS and PFS in patients aged 65 to 75 years (HR of 0.63 and 0.94, respectively), and in patients >75 years of age (OS HR: 0.90, PFS HR 0.97. Recently, the efficacy and safety of nivolumab were further reassessed in 70 patients ≥75 years (68 patients evaluable per response) with advanced non-squamous NSCLC. This subpopulation benefits from treatment with nivolumab, which reports a disease control rate of 42.9%, a median PFS and an OS of 3.2 and 7.6 months respectively.

### 3.3. Atezolizumab

Atezolizumab is an engineered anti PD-L1 monoclonal antibody that inhibits PD-L1/PD1 and PD-L1/B7.1 interaction. The activity of this antibody restores and improves the antitumor activity of T cells. Efficacy data of Atezolizumab have been evidenced by the phase III OAK study [60]. In this study, 1225 patients were recruited and divided into two groups. In the first group, Atezolizumab 1200 mg IV was given every 3 weeks, and Docetaxel 75 mg/m^2^ was given every 3 weeks in the second group. In Atezolizumab group, an improvement in OS compared with docetaxel was observed in ITT population: median overall survival was 13·8 months vs. 9·6 months; HR = 0.73, *p* = 0.0003). The improvement in observed OS was consistently demonstrated in the various patient subgroups, including those with brain metastases at baseline (HR of 0.54, 95% IC: 0.31–0.94, median OS of 20, 1 versus 11.9 months respectively with atezolizumab and docetaxel), without any history of smoking (HR = 0.71, 95% IC: 0.47–1.08, median OS of 16.3 versus 12, 6 months respectively with atezolizumab and docetaxel); furthermore overall survival improvement was similar in patients with squamous or non-squamous histology (HR = 0.73 in both groups). However, patients with EGFR mutations did not show any improvement in OS with atezolizumab compared to docetaxel. Patients in the subgroup with low or undetectable PD-L1 also demonstrated improved survival with atezolizumab (median OS 12.6 months vs. 8.9 months, HR 0.75, 95% CI 0.59–0.96). The survival advantage was persistent also in the subgroups divided according to age: the median OS was 13.2 and 10.5 months for 453 patients aged <65 years respectively treated with atezolizumab and docetaxel (HR 0.80). Median OS of 14.1 and 9.2 months for patients aged ≥65 years, respectively (HR 0.66). The percentage of elderly patients recruited in these studies were higher than others in the literature [61].

### 3.4. Durvalumab

Finally, in advanced NSCLC, a relevant activity coupled with a satisfactory safety profile has been proved for Durvalumab, a monoclonal IgG1κ antibody against PD-L1. In a randomized, double-blind, phase III study including patients with stage III locally advanced unresectable NSCLC who had not progressed following definitive platinum-based chemoradiation (at least 2 cycles), Durvalumab was compared to placebo and showed superior outcomes with median PFS was 16.8 months vs. 5.6 months (hazard ratio for PD or death = 0.52; 95%; *p* < 0.001). Furthermore, the response rate was 28.4% in durvalumab-treated patients vs. 16.0% in placebo-treated ones (*p* < 0.001) and the median time to death or distant metastasis favored durvalumab over placebo: 23.2 months vs. 14.6 months (*p* < 0.001). However, regarding OS, results favored durvalumab only in patients with PD-L1 ≥1%. The median age in the durvalumab group was 64 years (range 31–84). The PFS advantage within patients treated with durvalumab was observed in all groups regardless PD-L1 expression, though in the younger group (<65 years old) the benefit was higher (HR 0.43 95%IC 0.32–0.57) versus patients aged ≥ 65 years old (HR 0.74 95%IC 0.54–1.01) [62].

### 3.5. Recent Evidences in the Elderly Population

A systematic review, although not specific to NSCLC, compared the activity of ICIs in both young and elderly patients [63]. Nine randomized controlled trials (RCTs) of ICIs (ipilimumab, tremelimumab, nivolumab and pembrolizumab) were evaluated, including 5265 patients, divided using a variable cut off of 65 or 70 years depending on the study considered. The results showed an improvement in the OS in both groups, compared to standard chemotherapy (young patients: HR, 0.75; 95% CI, 0.68–0.82; older patients HR, 0.73; 95% CI, 0.62–0.87). Also, PFS analysis showed an improvement in both groups of patients (young patients: HR, 0.58; 95% CI, 0.40–0.84; elderly patients: HR, 0.77; 95% CI, 0.58–1.01). Accordingly, it appears that the elderly can derive significant benefit from the use of ICIs. Similar results emerged from another metanalysis which included nine RCTs (five comprising NSCLC patients); PD-1/PD-L1 inhibitors resulted in similar outcomes between adults younger vs. older than 65 years for OS [HR 0.68 (CI 0.61–0.75) vs. 0.64 (CI 0.54–0.76)] and PFS [HR 0.73 (CI 0.61–0.88) vs. 0.74 (CI 0.60–0.92)] [64]. Wu et al., using a 65 year cut-off, reported the benefit of immunotherapy treatment (anti CTLA-4 and anti PD-1/PD-L1 antibodies) in older subjects with different solid malignancies, achieving better outcomes than younger counterparts [65]. The reported systematic reviews were not focused on NSCLC and no data about durvalumab have been included. In a recent single center retrospective study in NSCLC patients, Lichtenstein et al. evaluated patients with NSCLC who initiated PD-1 and PD-L1 inhibitors and reported the main clinical outcomes. The authors originally described a not linear relationship in PFS with a numerical tendency towards longer PFS with advancing patient age with a peak in patient group aged 70–79 years. However, the oldest patients (more than 80 years) experienced shorter PFS compared with other age groups [66].

## 4. Comorbidities and Safety Profile

Although ICIs have a safe toxicity profile in NSCLC, data regarding toxicity in elderly population of these molecules is limited because most ICIs studies have involved a low number of elderly patients. Immune-related adverse events (irAEs) are defined as idiosyncratic adverse events to ICIs and may be more challenging in elderly patients due to reduced functional reserve, age-associated comorbidities and polypharmacy. In addition immunosenescence may play a central role in irAEs and researchers have shown that higher concentrations of inflammatory cytokines along with higher prevalence of autoantibodies, exclusively of IgG isotype, result in immune related toxicity including colitis, pneumonitis, hepatitis, nephritis and endocrinopathies [67]. The safety profile of nivolumab for patients with advanced, refractory, squamous non-small-cell lung cancer was primarly assessed by Rizvi et al. 17% of 117 patients reported grade 3–4 treatment-related adverse events, including: fatigue (4%), pneumonitis (3%), and diarrhoea (3%) [68]. Most frequent irAEs reported were cutaneous (15%), gastrointestinal (10%), endocrine (5%) and pulmonary (5%) manifestations. However, in this study, 86% patients were aged <75 years and only 14% were ≥ 75 years old. Furthermore, the toxicity results reported were not stratified for age. In a pooled analysis of 1030 patients (Renal Cell Carcinoma, Melanoma and NSCLC) treated with Nivolumab, toxicity was reported separately for three age groups (<65 years, 616 patients; 65 to <70 years, 414 patients; and ≥70 years, 212 patients). The incidence of any grade 1–2 and grade 3–4 toxicities were, respectively, 39% and 44% for patients less than 65 years; 35% and 45% for patients between 65 and 70 years; 37% and 46% for patients older than 70 years [69]. Most irAEs are generally mild and can be treated symptomatically.

In Keynote-010 grade 3–5 treatment-related adverse events were less common with pembrolizumab than with docetaxel. The most frequently reported treatment related adverse events of any grade in the pembrolizumab groups were decrease appetite (14%), fatigue (14%), nausea (11%), rash (9%), diarrhoea (7%) and asthenia (6%). Immune-related events, occurred in 20% of patients in the pembrolizumab 2 mg/kg group and in 19% of patients in the pembrolizumab 10 mg/kg group. The most common were hypothyroidism, hyperthyroidism, and pneumonitis. Three deaths were reported among patients treated with pembrolizumab due to immune adverse events [57]. Safety profiles were similar between younger and older participants.

ICIs in the elderly population resulted in a limited toxicity compared to standard chemotherapy as these agents are metabolized to peptides and aminoacids by circulating phagocytic cells and not by cytochrome P450 enzymes. Therefore, enzymatic competition is not expected; renal or hepatic dysfunction should have minimal impact on drug levels. As a consequence, no dose adjustment is recommended for patients with mild or moderate renal impairment (i.e., ≥30 mL/min creatinine clearance) or mild hepatic impairment [70]. Further studies in elderly NSCLC patients are required to systematically define the safely profile in this group of patients.

## 5. Gut microbiome and Immunosenescence: Implications for ICI’s

The role of the microbiome on innate and adaptive immune response and its contribution to homeostasis of human host is currently under investigation. The human gut microbiome is in a constant state of development from early to late stages of life. Research indicates that the gut microbiota composition is influenced by aging; in the elderly subject a reduction of commensal microorganisms and increase in levels of opportunists have been reported [71] possibly favoring the onset of chronic metabolic disorders such as diabetes, obesity, cardiovascular disease, and neurodegenerative disorders [72]. Claesson et al. documented a shift toward a *Clostridium* dominated microbiome with a significative prevalence of *Sporobacter*, *Faecalibacterium* and *Ruminococcus* species in elderly subjects [71]. Furthermore, microbiota modifications could contribute to frailty in older subjects as increased levels of *Bacteroidetes* commensals, *Oscillibacter*, *Alistipete* genera and *Enterobacteriaceae* family coupled with a progressive loss of *Lactobacillus* have been reported in frail elderly subjects when compared to healthy counterpart [73,74,75]. Preclinical metagenomic data postulated that gut microbiome could play a major role in modulating the tumor responses to both chemotherapeutic agents and PD-1/CTLA-4 based immunotherapy. Reduced intestinal motility and intestinal dysfunction promotes the loss of microbial diversity (dysbiosis) impairing the antigen presentation resulting in improved effector T cell function in the periphery and the tumor microenvironment [76]. In an elegant research, Gopalakrishnan et al. described that melanoma patients with a high diversity in the fecal microbiome experienced prolonged PFS compared to those with intermediate or low diversity (p=0.02 and 0.04, respectively) [76]. Similarly, in a preclinical research, mice with favorable microbiota composition exhibited better therapeutic activity of anti–PD-L1 treatment than unfavorable microbiome counterpart [77]. Matson et al. described the differences in microbiota between responders and non-responders among melanoma patients treated with monoclonal antibodies targeting PD-1 or CTLA4. Clustering analysis revealed more abundant levels of *Enterococcus faecium*, *Collinsella aerofaciens*, *Bifidobacterium adolescentis*, *Klebsiella pneumoniae*, *Veillonella parvula*, *Parabacteroides merdae*, *Lactobacillus* sp., and *Bifidobacterium longum* in responder patients whereas *Ruminococcus obeum* and *Roseburia intestinalis* were observed more frequently in non-responders. Thus, after inoculation with human commensal microbes of responder patients in mice, the tumor microenvironment showed an increase of *SIY-specific CD8+* T cells, but not of *FoxP3+CD4+* regulatory T cells which lead to an increased priming of tumor antigen–specific CD8+ T cells [78]. The microbiota-dependent immunostimulatory effects in patients treated with CTLA-4 and PD1/PD-L1 blockade may depend on the *CD11b+* dendritic cells mobilization of the lamina propria which promotes the Th-1 response against the Bacteroides fragilis capsular polysaccharides [79]. A differential expression in effector *CD4+*, *CD8+ T cells* and levels of regulatory T cells (*Treg*) and *MDSC*’s in relation to different microbiome exposure has also been postulated to explain the response to ICI’s. Finally, immunomodulation may also be influenced by polyunsaturated fatty acids which exhibit immune-stimulating effects in both the humoral and cellular immune systems. B-cell cytokine levels, including TNFα and IL-10, were found to be higher in obese subjects after eicosapentaenoic and docosahexaenoic acid supplementation [80,81].

## 6. Conclusions

The identification of immune checkpoint inhibitors has revolutionized the landscape of NSCLC. PD-1 and PD-L1 inhibitors, enhancing the T-cell response, are able to limit the immune escape phenomenon thus interfering with cancer cell progression. However, the promising results of ICIs in clinical studies in NSCLC should be carefully interpreted regarding the role of anti-PD1/PD-L1 agents among older subjects, as the phase III clinical trials may not allow definitive conclusions to be drawn. Elderly patients recruited in clinical trials are often very limited in number and may not be representative of the elderly population with NSCLC requiring treatment in clinical practice. Recent studies have provided novel findings about the efficacy and safety of monoclonal antibodies targeting PD1/PD-L1 axis and CTLA-4 in particular in patients aged <75 years; conversely, in older subjects, results are more heterogeneous. In this group of patients, metagenomic data about gut microbiome and the influences on T Cell regulatory pathways may offer novel insights in understanding the mechanisms of impaired response to ICIs. Further research is required to define the magnitude of benefits in this important subset of patients.

## Figures and Tables

**Figure 1 ijms-20-02258-f001:**
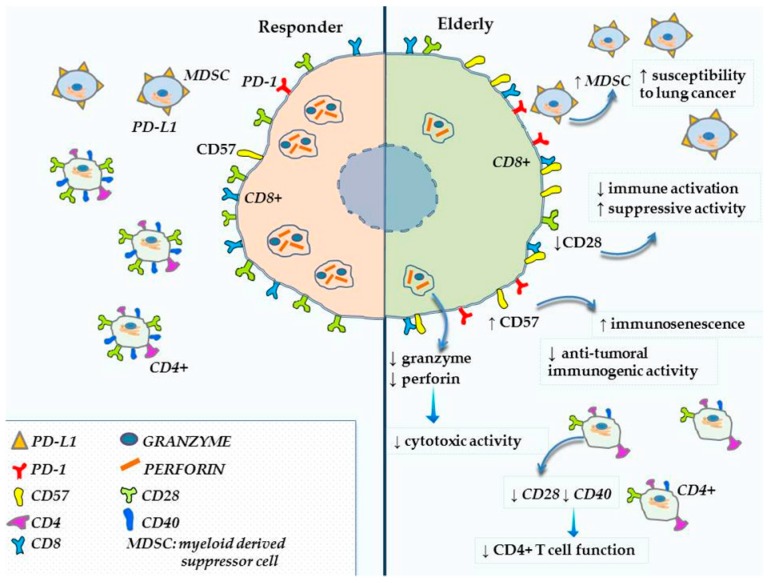
Age dependent dynamic T cell regulation influences immune checkpoints.

**Table 1 ijms-20-02258-t001:** Results of Immune checkpoint inhibitors in clinical trials adjusted by age.

MoAbs Targeting PD-1/PD-L1	Clinical Trial	Target Population	OS (HR 95%IC)	Median Age in Treated Arm (Range)
Nivolumab monotherapy	CheckMate017	Squamous NSCLC		62.0 (39–85)
<65 yr			0.62 (0.44–0.89)	
≥65 <75 yr			0.51 (0.32–0.82)	
≥75 yr			1.76 (0.77–4.05)	
Nivolumab monotherapy	CheckMate057	Nonsquamous NSCLC		61.0 (37–84)
<65 yr			0.81 (0.62–1.04)	
≥65 <75 yr			0.63 (0.45–0.89)	
≥75 yr			0.90 (0.43–1.87)	
Pembrolizumab monotherapy	KEYNOTE-024	Nonsquamous and squamous NSCLC		64.5 (33–90)
< 65 yr			0.61 (0.40–0.92)	
≥65 yr			0.45 (0.29–0.70)	
Pembrolizumab monotherapy	KEYNOTE-010	Nonsquamous and squamous NSCLC		63.0 (56–69)
< 65 yr			0·63 (0·50–0·79)	
≥65 yr			0·76 (0·57–1·02)	
Atezolizumab monotherapy	OAK	Nonsquamous and squamous NSCLC		63.0 (33–82)
< 65 yr			0·80 (0·64–1·00)	
≥65 yr			0·66 (0·52–0·83)	
Durvalumab monotherapy	PACIFIC	Nonsquamous and squamous NSCLC		64.0 (31–84)
< 65 yr			0.62 (0.44–0.86)	
≥65 yr			0.76 (0.55–1.06)	

MoAbs: Monoclonal Antibodies; NSCLC: Non-Small Cell Lung Cancer; PD-1: Programmed Death; PD-L1: Programmed Death Ligand.

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
