# Peer review of "Immune Checkpoint Blockade for Advanced NSCLC: A New Landscape for Elderly Patients"

_ijms, 2019, doi:10.3390/ijms20092258_

Round 1
Reviewer 1 Report
The article presents a good view on the potential use of immune checkpoint inhibitors for elderly patients with advanced NSCLC. The authors present a strong case based on immunosenescense and supported with past studies and references. However, the article can be improved in light of new studies involving immune checkpoint inhibitors that may also be related to immunosenescence.
Recent clinical studies with immune checkpoint inhibitors indicate that its success is dependent on an individual's microbiome (ref 1-3). There has been association of the microbiome with immunosenescence (4). Furthermore, the role of omega 3 fatty acids in immunosenescence should be considered. Finally, a recent hypothesis (5) on how omega 3 fatty acids, the gut microbiome and immune system interact in regulating immune checkpoint inhibition may be worth addressing especially in light of immunosenescence.
I suggest incorporating an additional section addressing the aforementioned factors and how these may be considered for elderly patients.
A minor editorial comment is that the sections are too long (e.g. Section 3 and 4) which makes it a bit tedious to read. I would suggest that subsections be made especially when describing each ICI antibody and clinical study.
References:
1. Routy, B. et al. Gut microbiome influences efficacy of PD-1-based immunotherapy against epithelial tumors. Science 2018, 359, 91-97,10.1126/science.aan3706.
2. Gopalakrishnan, V et al. Gut microbiome modulates response to anti-PD-1 immunotherapy in melanoma patients. Science 2018 359,97-103. doi: 10.1126/science.aan4236.
3. Matson, V. et al. The commensal microbiome is associated with anti-PD-1 efficacy in metastatic melanoma patients. Science 2018 359,104-108, doi: 10.1126/science.aao3290.
4. Amsterdam D and Ostrov BE. Impact of the microbiome on Immunosenscence. Immunological Investigations 2018 47,801-811 https://doi.org/10.1080/08820139.2018.1537570
5. Ilag LL. Are Long-Chain Polyunsaturated Fatty Acids the Link between the Immune System and the Microbiome towards modulating cancer? Medicines 2018 5(3),102, https://doi.org/10.3390/medicines5030102
Author Response
Recent clinical studies with immune checkpoint inhibitors indicate that its success is dependent on an individual's microbiome (ref 1-3). There has been association of the microbiome with immunosenescence (4). Furthermore, the role of omega 3 fatty acids in immunosenescence should be considered. Finally, a recent hypothesis (5) on how omega 3 fatty acids, the gut microbiome and immune system interact in regulating immune checkpoint inhibition may be worth addressing especially in light of immunosenescence. I suggest incorporating an additional section addressing the aforementioned factors and how these may be considered for elderly patients. A minor editorial comment is that the sections are too long (e.g. Section 3 and 4) which makes it a bit tedious to read.
I would suggest that subsections be made especially when describing each ICI antibody and clinical study. • We wish to thank the Reviewer. According your indication we have added a paragraph discussing the more recent advances in this field. We discussed the impact of microbiome on immunosenescense and its implication on response to ICI’s. This topic is original, relevant and not fully explored; we agree that it could be appealing for the readers. The paragraph on clinical trials have been divided in sub-parapraphs. Redundant parts have been removed.
Reviewer 2 Report
Better understanding of the efficacy and toxicity of immunotherapy in the elderly population is an open issue in the field. In this manuscript, the authors summarize results of clinical trials in NSCLC, with a focus on survival results within older age subgroups of each paper. However, there are already similar recent papers such as
1) Elias et al, Efficacy of PD-1 & PD-L1 inhibitors in older adults: a meta-analysis. (PMID 29618381)
2) Wu et al, Correlation between patients' age and cancer immunotherapy efficacy. (PMID 29898988)
3) Lichtenstein et al, Impact of Age on Outcomes with Immunotherapy in Patients with Non-Small Cell Lung Cancer (PMID: 30476576)
These papers are not cited in this paper, but have conducted more detailed meta-analysis including survival analysis and forest plots. I'm not sure what more information this review provides. Yes, some studies (including the study the authors cited) include all cancer types and do not focus on NSCLC, but the authors should provide reasoning for why immunotherapy response in the .elderly could be different in NSCLC compared to other cancer types.
Overall, the authors should be referencing these studies and emphasizing novel data or viewpoints included in their review.
Author Response
Better understanding of the efficacy and toxicity of immunotherapy in the elderly population is an open issue in the field. In this manuscript, the authors summarize results of clinical trials in NSCLC, with a focus on survival results within older age subgroups of each paper. However, there are already similar recent papers such as Elias et al, Efficacy of PD-1 & PD-L1 inhibitors in older adults: a meta-analysis. (PMID 29618381) Wu et al, Correlation between patients' age and cancer immunotherapy efficacy. (PMID 29898988) Lichtenstein et al, Impact of Age on Outcomes with Immunotherapy in Patients with Non-Small Cell Lung Cancer (PMID: 30476576). These papers are not cited in this paper, but have conducted more detailed meta-analysis including survival analysis and forest plots. I'm not sure what more information this review provides. Yes, some studies (including the study the authors cited) include all cancer types and do not focus on NSCLC, but the authors should provide reasoning for why immunotherapy response in the .elderly could be different in NSCLC compared to other cancer types. Overall, the authors should be referencing these studies and emphasizing novel data or viewpoints included in their review. •
We wish to thank the Reviewer for highlighting this criticism. The role of immunosenescence and response to ICI’s is of great relevance as older subjects are generally underrepresented in clinical trials. The three important papers suggested are now discussed in the revised version of the manuscript. The two metanalysis suggested present well organised and comprehensive data on age related response to ICIs; on the other hand, they did not include only NSCLC patients and in this scenario, we believe that our manuscript adds novel insights in the treatment of NSCLC. Furthermore, the literature in this field is under continue evolution and to best of our knowledge no data on Durvalumab are reported in the presented systematic reviews. We have added - highlighted in the text – the key points for considering ICI’s in the elderly as an innovative topic with a particular relevance. The novel paragraph about the impact of human microbiota on response to ICI’s and the network with immunosenescence and T cell regulation contributes to originality of the paper.
Round 2
Reviewer 2 Report
The authors have made an effort to incorporate some newer studies and a new section about gut microbiome. While they mention that "Unfortunately, all the reported systematic reviews were not focused on NSCLC", they did not provide reasoning for why we might . expect immunotherapy response in the elderly to differ in NSCLC compared to other cancer types. Thus, I am still not sure why the specific focus of the paper on NSCLC, given that immunotherapies are approved across many tumor types.
I appreciate the addition of Table 1, which summarizes the trials and drugs, but can column(s) be added that relate this table to response in elderly subpopulation vs others? This would allow the authors themselves, as well as other readers, to compare the efficacy of these drugs more easily. Otherwise this table is not very useful given that the median age in all these trials are similar (they are all lung cancer after all). The table is also barely introduced in the main text.
I like the addition of the section on gut microbiome. The authors mention that elderly have lower gut microbiota diversity, and also that current studies suggest that increased diversity leads to better immunotherapy response. However, these two should be linked together to summarize the section! What is specifically known about the presence of Ruminococcus/Bacteroidales/other levels in elderly? Do the authors think that elderly may respond worse to immunotherapy because of their lower gut microbiota diversity? Many of the immunotherapy drugs in the clinical trials section seem to work equally well in older and younger patients, which seems to contradict this data. Logically, I think this section should come after the clinical trials section.
The conclusion could expand a little more on the "the promising results of ICIs." Did any dt work better or worse in elderly patients? What is the conclusion of the gut microbiota section?
Author Response
Immune Checkpoint Blockade for Advanced NSCLC: A New Landscape for Elderly Patients
Perrotta et al.
Point by point response.
The authors have made an effort to incorporate some newer studies and a new section about gut microbiome. While they mention that "Unfortunately, all the reported systematic reviews were not focused on NSCLC", they did not provide reasoning for why we might . expect immunotherapy response in the elderly to differ in NSCLC compared to other cancer types. Thus, I am still not sure why the specific focus of the paper on NSCLC, given that immunotherapies are approved across many tumor types.
We really thank the reviewer for arising this criticism. In this review we examined the current approved ICIs for advanced NSCLC. Though the promising result of immunotherapy are generalisable to other cancer types, the drugs related outcomes are different in various cancers (i.e. some agents are approved from some cancer types and not for others). In this review, we focused on the ‘state of art’ therapy for advanced NSCLC with the aim to highlight the results in the elderly population. We strongly believe that this topic could be suitable for pulmonologist and oncologist readers. However, we have removed the term ‘Unfortunately’ which may be misleading.
I appreciate the addition of Table 1, which summarizes the trials and drugs, but can column(s) be added that relate this table to response in elderly subpopulation vs others? This would allow the authors themselves, as well as other readers, to compare the efficacy of these drugs more easily. Otherwise this table is not very useful given that the median age in all these trials are similar (they are all lung cancer after all). The table is also barely introduced in the main text.
We thank the reviewer. We add the OS for the different age groups; this allows a quick overview about the clinical trial results. The Table is now adequately presented in the main text.
I like the addition of the section on gut microbiome. The authors mention that elderly have lower gut microbiota diversity, and also that current studies suggest that increased diversity leads to better immunotherapy response. However, these two should be linked together to summarize the section! What is specifically known about the presence of Ruminococcus/Bacteroidales/other levels in elderly? Do the authors think that elderly may respond worse to immunotherapy because of their lower gut microbiota diversity? Many of the immunotherapy drugs in the clinical trials section seem to work equally well in older and younger patients, which seems to contradict this data. Logically, I think this section should come after the clinical trials section.
We thank the reviewer and according the indications we have moved the ‘Gut Microbiome’ paragraph after the discussion about ICIs. The gut microbiota paragraph has been remodulated according your indications. We have included the evidences about the microbiota composition in elderly subjects. We have better discussed and presented the current knowledge about the microbiota diversity and response to ICIs.
The conclusion could expand a little more on the "the promising results of ICIs." Did any dt work better or worse in elderly patients? What is the conclusion of the gut microbiota section?
In the conclusion paragraph we have now incorporated the results of the microbiome section. We have highlighted that in patients older than 75 years clinical results are more heterogenous than younger counterpart (Checkmate 017 and Lichtenstein, M.R.L.; Nipp, R.D.; Muzikansky, A.; Goodwin, K.; Anderson, D.; Newcomb, R.A.; Gainor, J.F. Impact of Age on Outcomes with Immunotherapy in Patients with Non-Small Cell Lung Cancer. J. Thorac. Oncol. 2019, 14, 547–552.) In this view, the microbiota composition, the T cell regulatory pathways may be the substrate of a limited efficacy.
